# Evidence of Immune Modulators in the Secretome of the Equine Tapeworm *Anoplocephala perfoliata*

**DOI:** 10.3390/pathogens10070912

**Published:** 2021-07-20

**Authors:** Boontarikaan Wititkornkul, Benjamin J. Hulme, John J. Tomes, Nathan R. Allen, Chelsea N. Davis, Sarah D. Davey, Alan R. Cookson, Helen C. Phillips, Matthew J. Hegarty, Martin T. Swain, Peter M. Brophy, Ruth E. Wonfor, Russell M. Morphew

**Affiliations:** 1Institute of Biological, Environmental and Rural Sciences, Aberystwyth University, Aberystwyth SY23 3DA, UK; bow3@aber.ac.uk (B.W.); beh23@aber.ac.uk (B.J.H.); jjt12@aber.ac.uk (J.J.T.); nra3@aber.ac.uk (N.R.A.); chd31@aber.ac.uk (C.N.D.); sad35@aber.ac.uk (S.D.D.); akc@aber.ac.uk (A.R.C.); hcp5@aber.ac.uk (H.C.P.); ayh@aber.ac.uk (M.J.H.); mts11@aber.ac.uk (M.T.S.); pmb@aber.ac.uk (P.M.B.); 2Faculty of Veterinary Science, Rajamangala University of Technology Srivijaya, Nakhon Si Thammarat 80240, Thailand

**Keywords:** *Anoplocephala perfoliata*, transcriptome, secretome, extracellular vesicles, EV surface, EV depleted ESP, parasite–host interaction

## Abstract

*Anoplocephala perfoliata* is a neglected gastro-intestinal tapeworm, commonly infecting horses worldwide. Molecular investigation of *A. perfoliata* is hampered by a lack of tools to better understand the host–parasite interface. This interface is likely influenced by parasite derived immune modulators released in the secretome as free proteins or components of extracellular vesicles (EVs). Therefore, adult RNA was sequenced and de novo assembled to generate the first *A. perfoliata* transcriptome. In addition, excretory secretory products (ESP) from adult *A. perfoliata* were collected and EVs isolated using size exclusion chromatography, prior to proteomic analysis of the EVs, the EV surface and EV depleted ESP. Transcriptome analysis revealed 454 sequences homologous to known helminth immune modulators including two novel Sigma class GSTs, five α-HSP90s, and three α-enolases with isoforms of all three observed within the proteomic analysis of the secretome. Furthermore, secretome proteomics identified common helminth proteins across each sample with known EV markers, such as annexins and tetraspanins, observed in EV fractions. Importantly, 49 of the 454 putative immune modulators were identified across the secretome proteomics contained within and on the surface of EVs in addition to those identified in free ESP. This work provides the molecular tools for *A. perfoliata* to reveal key players in the host–parasite interaction within the horse host.

## 1. Introduction

The gastro-intestinal tapeworm, *Anoplocephala perfoliata*, is one of the most prevalent tapeworm species that infects horses worldwide, with prevalence estimated between 15.8–44% of horses, yet it remains a neglected parasite with respect to management as low infection rates are often asymptomatic [1,2,3,4,5,6]. However, high burdens of *A. perfoliata* are linked to abdominal disturbance or pain, including spasmodic colic, ileal impaction [7], ileocaecal or caeco–caecal intussusception [8,9] and ileal or caecal rupture [10,11] due to the accumulation of adults at the ileocaecal region that attach to the caecal epithelium [2,12,13,14,15]. In such high-level infections, a localised caecal mucosal inflammatory response is activated at the site of the tapeworm attachment [2,16,17,18]. Severe caecal tissue damage and dysfunction likely predisposes horses to colic, which is often fatal during later stages without treatment [2,14]. Such documented evidence demonstrates the importance of further understanding the pathology and biology of *A. perfoliata* for future diagnostics and control options.

At present, there is limited mechanistic evidence of how *A. perfoliata* interacts with the horse intestinal environment. Therefore, an increased understanding of the underpinning molecular biology of *A. perfoliata* is required to delineate parasite–host interactions. Polyomic technologies, such as transcriptomics and proteomics, have generated new omic database resources to support future control for several other important helminths [19,20,21,22,23]. To date, molecular analysis of *A. perfoliata* has focussed on the mitochondrial genome sequence in order to simply assess the phylogenetic relationship with the closely related *Anoplocephala magna* [22,24]. However, at present, there is no reference genome or supporting transcriptome profiles to support discovery biology in *A. perfoliata*, which precludes in depth proteomic profiling. Furthermore, comprehensive nucleotide support would provide unique biological information for *A. perfoliata*, to facilitate the identification of genes likely involved in the pathogenesis of tapeworm infections and tapeworm-host interactions.

Molecules secreted by helminth parasites into the host environment during the course of infection are termed the excretory/secretory products (ESP) or the secretome and contain a variety of soluble proteins, glycoproteins, carbohydrates, lipids, and metabolites many of which are known to have an important role in helminth-mediated immunomodulation [16,25,26,27,28,29,30,31,32,33,34]. Investigation of the ESP from tapeworms, such as the protoscoleces of *Echinococcus granulosus*, have demonstrated regulation of immune cell differentiation, such as B10, B17, and Th17 cells in infected mice, accompanied by a downregulation of the inflammatory response [32]. Furthermore, *A. perfoliata* has been suggested to downregulate T-cell responses in the horse in live infections, which was partly attributed to ESP, which inhibited growth and induced apoptosis of Jurkat cells (human T-cell line) in vitro [16]. However, at present the active key component(s) in *A. perfoliata* ESP driving these potential host immunomodulatory mechanisms are yet to be determined [16].

Extracellular vesicles (EVs) are lipid membrane-bound structures released from helminths as part of the ESP. Parasite EVs are released into the host extracellular environment and are likely candidates for intercellular communication and immunomodulation [35,36,37]. To date, EVs released from tapeworms have been identified in adult *Hymenolepis diminuta* [38] and *Taenia asiatica* [39], and from metacestode stages of *Taenia pisiformis* [40], *Echinococcus granulosus* [41,42,43], *Taenia crassiceps*, *Mesocestoides corti*, and *Echinococcus multilocularis* [44]. However, EVs from adult equine tapeworms, including *A. perfoliata*, have not yet been identified.

Key secretory proteins involved in immune modulation, host interaction and parasite survival have been identified as part of the ESP as free proteins or as components of EVs for a number of helminths. Annexins, actins, cathepsin proteases, heat shock proteins (HSPs), helminth defense molecules (HDMs), glutathione transferases (GSTs), and fatty-acid binding proteins (FABP) are among a number of such proteins that have been identified from *Fasciola hepatica* [27,30,45,46,47,48], *Calicophoron daubneyi* (Rumen Fluke) [21,49], *Schistosoma*
*japonicum* [50], and *Schistosoma mansoni* [51] representing well characterised platyhelminths. Thus, in depth characterisation of helminth secretomes has the potential to uncover host parasite interaction mechanisms.

Therefore, understanding of host–parasite interactions and subsequent pathogenesis are hampered in research neglected parasites such as *A. perfoliata* by a lack of fundamental molecular resources. Thus, this work generated the first transcriptome for adult *A. perfoliata* to support proteomic investigations. In addition, deploying this novel transcriptome revealed the *A. perfoliata* secretome, including EVs, EV surface expressed proteins, and EV depleted ESP via GeLC and Gel free proteomic approaches. This first comprehensive coverage of the *A. perfoliata* secretome has revealed the likely key players in the host–parasite interaction within the horse host.

## 2. Results

### 2.1. De Novo Assembly of the A. perfoliata Transcriptome

RNA sequencing across all six samples resulted in a total of 109,267,236 paired-end reads. The trimmed reads underwent de novo assembly individually and a total of 199,943 transcripts sequences were obtained. A total of 2353 sequences were determined to be host (*E. caballus*) when the peptide candidates were blasted against the *E. caballus* genome and related flatworm genome (*Hymenolepis microstoma*). Following removal of homologous sequences and host contamination, a total of 74,607 remained, with the total assembled contigs length of 56,913,324 bp and mean ± SD contig length of 763 ± 762 bp. Summary statistics for the *A. perfoliata* de novo transcriptome assembly are presented in Table 1. The assembled transcriptome was converted to protein sequences, using best predicted coding regions, resulting in 34,341 protein sequences (Table 1).

### 2.2. Transcriptome Functional Annotation and Gene Ontology Terms Analysis

A total of 19,445 (56.6%) sequences were successfully annotated to the top three species hits to related tapeworm species including *Hymenolepis diminuta*, *Echinococcus granulosus*, and *Hymenolepis microstoma*. However, Omicsbox functional annotation revealed that 3244 (9.4%) protein sequences could not be annotated. A further 6199 (18.1%) sequences were annotated with Blast Hits and 5453 (15.9%) with GO term mapping. The majority of GO terms were classified according to the three main GO categories; biological processes (33.5%), molecular functions (39.7%), and cellular components (26.8%). The most frequent GO terms identified by level 3 are demonstrated in Appendix A. Protein descriptions were assigned to 27,950 (81.39%) of the transcript sequences, whereas 6391 (18.61%) sequences were unnamed protein products or uncharacterised proteins; 1738 (5.06%) were of unknown function; 3244 (9.45%) did not have information provided and 1409 (4.10%) were hypothetical proteins/transcripts.

### 2.3. Transcripts Expression of A. perfoliata Transcriptome

The mean ± SD TPM was 12.4 ± 222, with the top 50 most abundant transcripts summarised in Table 2. Known genes of interest in other platyhelminths were noted amongst the top 50 abundance transcripts, such as, Dynein light chain (IPR037177) superfamily, EF-hand domain pair (IPR011992), Armadillo-like helical (IPR011989), Armadillo-type fold (IPR016024), and profilin superfamily (IPR036140). Protein families and domains of highly expressed genes related to these superfamilies were also found, similar to other platyhelminths, such as Dynein light chain, type 1/2 family (IPR001372), profilin family (IPR005455), EF-hand domain (IPR002048), and domain of unknown function, DUF5734 (IPR043792). Thus, demonstrating the likely validity of the transcript dataset for *A. perfoliata*.

### 2.4. Bioinformatics of Potential Immune Modulators

An initial *A. perfoliata* transcriptome analysis by tBLASTn and PFAM was performed using 73 known immune modulators from helminths. This analysis demonstrated significant hits for 43 of the 73 bait immune modulators leading to a total of 454 unique contigs which were identified as potential immune modulator homologs (cutoff 1 × 10^−15^) (Appendix A). The top five hits, or less is fewer than five hits returned, for each of the 43 bait proteins returning hits were confirmed as homologs using Pfam domain searches with over 70% domain conservation and provided a total of 83 unique contigs confirmed as potential immune modulators.

#### 2.4.1. Characterisation of Novel *A. perfoliata* Sigma Class GSTs

A total of 4 *A. perfoliata* sequences were identified by tBLASTn as potential Sigma class GST protein sequences, homologous to protein sequences of recognised Sigma class GSTs from mammals and helminths (cutoff 1 × 10^−15^). Subsequently, InterProScan followed by manual BLASTp against the NCBI (nr) database confirmed that three of these sequences were potential Sigma class GST *A. perfoliata* sequences (GST superfamily IPR040079) representing two distinct enzymes (ApGST-S1, ApGST-S2.1, and ApGST-S2.2 where 2.1 and 2.2 share 100% amino-acid sequence identity).

The secondary characteristic structure of multiple aligned *A. perfoliata* Sigma GST homolog sequences were predicted, with three β-strands and nine α-helices demonstrating the consistency of the secondary characteristic structure between ApGST-S1, ApGST-S2 (2.1 and 2.2), and recognised Sigma class GST sequences (Appendix A). Sigma class GST homologs were investigated based on the GSH-binding sites in the N-terminal domain such as the catalytic tyrosine residue at the end of the first β-strand (Tyr^8^), Phe^9^, Arg^14^, Trp^39^, Lys^43^, Pro^52^, and Ser^64^) and the substrate binding sites in the C-terminal domain [52,53,54,55]. Only ApGST-S2.1 and 2.2 (212 amino acids) contained the highly conserved tyrosine residue (Tyr^8^) at the end of the first β-strand and also demonstrated a high homology to other GSH-binding sites (Appendix A).

A maximum likelihood (ML) phylogenetic tree generated from the 3 *A. perfoliata* and 16 recognised Sigma class GST sequences demonstrated that ApGST-S1, ApGST-S2.1, and ApGST-S2.2 were clustered in a Cestode clade, which suggested that all three should be included in the Sigma class of the GST superfamily (Figure 1). ApGST-S1 clustered closest to *H. microstoma* Sigma like GST (accession CDS25704) with a bootstrap value of 52%, and also clustered in the Sigma group of *Echinococcus multicularis* Sigma like GST (accession CDS39356) and *Echinococcus granulosus* isozyme (accession EUB60467) with a high bootstrap value (100%) (Figure 1).

#### 2.4.2. Characterisation of Novel *A. perfoliata* Heat Shock Protein 90

A total of nine *A. perfoliata* sequences were identified by tBLASTn as potential HSP90 protein sequences, homologous to protein sequences of recognised HSP90 family including alpha and beta isoforms from mammals and helminths (cutoff 1 × 10^−20^). The InterProScan followed by manual BLASTp against NCBI (nr) database confirmed eight *A. perfoliata* sequences as part of the HSP90 protein family (IPR001404) representing five likely distinct *A. perfoliata* HSP90s.

The secondary characteristic structure of multiple aligned HSP90 alpha (HSP90α) *A. perfoliata* homologous sequences was predicted, with 17 β-strands and 25 α-helices demonstrating the consistency of the secondary characteristic structure between all eight novel *A. perfoliata* and recognised HSP90α sequences (Appendix A). Among the eight novel isoforms of *A. perfoliata* HSP90α sequences, ApHSP90-4 (743 amino acids) contained a unique motif; the MEEVD peptide sequence in the C terminal end of cytoplasmic HSP90 isoforms, whereas ApHSP90-5.1 (777 amino acids) and ApHSP90-5.2 (575 amino acids) contained KEEL peptide sequence 75% identical to KDEL peptide sequence of unique endoplasmic reticulum isoforms (human GRP94) [56]. The cytosolic HSP90α was investigated based on a signature sequence LIP and EDD peptide sequences at residues 80–82 and 701–703, respectively [57]. However, all *A. perfoliata* sequences lacked this signature as did the additional included cestode sequences.

The analysis of the evolutionary relationships between the 8 *A. perfoliata* HSP90 and 27 recognised HSP90 protein member sequences demonstrated that there were no *A. perfoliata* HSP90 sequence clustered within HSP90 beta (HSP90β) clades (Figure 2). All eight potential *A. perfoliata* HSP90 sequences were clustered in a branch of HSP90α into a Cestode and Trematode specific clade. Of which, ApHSP90-4 was in a HSP90α Cestode specific clade containing *E. granulosus* (accession XP_024345770.1 and CDI70178.1) with good bootstrap support (bootstrap value 97%). The phylogenetic analysis suggests five *A. perfoliata* HSP90s of which ApHSP90-1.1 to 2.2 and ApHSP90-3 were clustered into a Cestode specific clade of three recognised HSP90 *H. microstoma* (accession CDS28179.1), *E. granulosus* (accession CDS25067.1), and *E. multilocularis* (accession CDS39694.1) with good bootstrap support (Bootstrap value 75%) (Figure 2).

#### 2.4.3. Characterisation of Novel *A. perfoliata* Alpha-Enolase

A total of five *A. perfoliata* sequences were identified by tBLASTn as potential alpha-Enolase (α-Enolase) protein sequences, homologous to protein sequences of recognised α-Enolase from mammals and helminths (cutoff 1 × 10^−15^). Subsequently, InterProScan followed by manual BLASTp against the NCBI (nr) database confirmed that five of these sequences were potential α-Enolase *A. perfoliata* sequences (Enolase superfamily IPR000941). There were three *A. perfoliata* α-Enolase enzymes that retained all catalytic residues and, therefore, likely to be catalytically functional, whilst the remaining two sequences represented likely incomplete fragments.

The secondary characteristic structure of the three full length α-Enolase *A. perfoliata* homologous sequences were predicted. All three α-Enolase *A. perfoliata* sequences; Apα-Enolase-1 (433 amino acids), 2 (456 amino acids) and 3 (456 amino acids) showed comparable numbers of β-strands and α-helices when compared to human α-Enolase (11 β-strands and 16 α-helices) demonstrating the consistency of the secondary characteristic structure between the three novel α-Enolase from *A. perfoliata* and recognised α-Enolase sequences (Appendix A). All three α-Enolase *A. perfoliata* sequences and recognised α-Enolase sequences conserved amino acid residues imperative for proper catalytic function (His^158^, Glu^167^, Glu^210^, Lys^343^, Lys^394^, respective positions in human α-Enolase) [58].

The analysis of the evolutionary relationships between all 3 *A. perfoliata* α-Enolase and 14 recognised α-Enolase protein member sequences demonstrated that all 3 *A. perfoliata* α-Enolase sequences (Apα-Enolase-1, -2, and -3) clustered in a Cestode specific clade containing *H. microstoma* α-Enolase sequence (accession CDS26422 and CDS30005), *T. solium* (accession AHB59732), *E. multilocularis* (accession CDS37852), and *E. granulosus* (accession ACY30465) with good bootstrap support (Figure 3).

### 2.5. Morphological Characterisation and Size Distribution of A. perfoliata EVs

Both transmission electron microscopy (TEM) and nanoparticle tracking analysis (NTA) confirmed that *A. perfoliata* secreted EVs during in vitro maintenance (Figure 4a). TEM analysis demonstrated the morphological characteristics of size exclusion chromatography (SEC) purified *A. perfoliata* EVs were in a spherical shaped (cup-shaped) membrane surrounded by a phospholipid bilayer structure (Figure 4a). The proportion of the size distribution of SEC purified *A. perfoliata* exosome (30–100 nm) and microvesicles (100–1000 nm) determined by TEM was 86% and 14%, respectively (Figure 4b). NTA demonstrated that the majority of the EV population were found to be 67–213 nm (Figure 4c), with a mean concentration of 1.57 × 10^9^ EV particles/mL (Table 3). The mean estimated particle size (mean ± SD) of SEC purified *A. perfoliata* EVs (three replicates) measured by TEM and NTA were approximately 64.16 ± 28.50 nm (*n* = 200) and 199.1 ± 108.7 nm in diameter, respectively. TEM showed the greatest EV size was 214.57 nm whereas and the smallest EVs size was 30.17 nm in diameter. The summary statistics of NTA of SEC purified *A. perfoliata* EVs at 1:600 dilution (*n* = 3) are shown in Table 3.

### 2.6. Protein Profiling of A. perfoliata Proteomics Datasets

Three biological replicates of both whole EVs and EV depleted ESP produced similar patterns of protein bands for each separate sample demonstrating the similarity amongst biological replicates. Protein bands in *A. perfoliata* EV depleted ESP demonstrated more frequent and dense proteins on 1D SDS-PAGE gels compared to the whole EVs (Appendix A).

The resulting three *A. perfoliata* mass spectrometry datasets including whole EVs, EV surface and EV depleted ESP were analysed through MASCOT via MS/MS Ion Search against the *A. perfoliata* transcriptome for protein identification. The full list of proteins identified in each *A. perfoliata* proteomics datasets is available in Appendix A. A total of 315 proteins were identified from *A. perfoliata* whole EVs, 301 proteins from the EV surface and 596 proteins from EV deleted ESP (Figure 5).

A total of 107 proteins were common between all three proteomics datasets (Figure 5), with the most abundant proteins in all datasets being WD repeat and FYVE domain-containing protein 3 (Table 4, Table 5 and Table 6). A total of 142 identified proteins were common between whole EVs and the EV surface with a total of 474 proteins identified in or on EVs. Well-known identified EV markers of interest as defined by the Exocarta database [59,60] and Vesiclepedia data [61] in *A. perfoliata* EV and EV surface proteomics datasets were noted amongst the top 50 most abundant proteins, such as, annexin, actin, myosin, enolase, phosphoglycerate kinase, heat shock 70 kDa protein, molecular chaperone HtpG, and programmed cell death 6-interacting protein (Table 4 and Table 5). On the surface of *A. perfoliata* EVs, protein pumps and transporters were identified such as ATP binding cassette subfamily B (MDR:TAP), multidrug resistance protein, V type proton ATPase 116 kDa subunit A, plasma membrane calcium-transporting ATPase 3 and solute carrier family 5 (Table 5). Key secretory proteins linked to the host–parasite interface such as enolase and calpain were identified in the top 50 most abundant of *A. perfoliata* EV depleted ESP (Table 6).

The 454 putative immune modulator sequences that were identified in the transcript were also assessed in the *A. perfoliata* proteomics data, with a total of 49 identified as expressed proteins across all three datasets, including 22 expressed in EVs, 16 on the EV surface and 40 in the EV depleted ESP (Appendix A). Of note, only a single Sigma class GST was identified across the proteomic datasets. ApGST-S1 was identified within the EV depleted ESP proteomic dataset and relatively low abundance. When assessing the proteomics datasets for HSP90 and α-Enolase, one HSP90 (ApHSP90-4) and two α-Enolase (Apα-Enolase-1 and 2) were observed across all three proteomic datasets analysed; namely whole EVs, EV surface proteins and EV depleted ESP. ApHSP90-4 and Apα-Enolase-1 were extremely abundant in the analysis featuring in the top 30 of all three datasets (27th and 23rd in whole EVs, 30th and 24th on the EV surface and 29th and 3rd in the ESP, respectively; Table 4, Table 5 and Table 6). Additionally, Apα-Enolase-2 was identified within the EV depleted ESP proteomic dataset at very low abundance.

### 2.7. Gene Ontology Enrichment Analysis

A total of 173 GO terms were not propagated up the hierarchy (*p* < 0.05 identified significance), of which 45, 42, and 86 GO terms were enriched in whole EVs, the EV surface and EV depleted ESP, respectively. The comparison of GO term enrichment from all three proteomics datasets is presented in Appendix A. A total of 20 GO terms were enriched across all secretome proteomic datasets, with most GO terms being in the biological processes group, of which calcium ion binding was found to be the most enriched, followed by Arp2/3 complex-mediated actin nucleation, microtubule-based process and gluconeogenesis. Seven GO terms enriched in EV samples only were all categorised in biological processes, with carbohydrate transmembrane transport being the most enriched followed by transmembrane transport, inorganic anion transport, peptidyl-lysine modification to peptidyl-hypusine, calcium-mediated signaling, protein phosphorylation, and protein processing. In the EV surface protein samples, the three most enriched GO terms were categorised in biological processes including bile acid and bile salt transport, seryl-tRNA aminoacylation and urea cycle whereas the three enriched GO terms categorised in cellular component categories included membrane, virion, and nascent polypeptide-associated complex. The top five most enriched GO terms in EV depleted ESP were categorised in biological processes included carbohydrate metabolic process, formaldehyde catabolic process, negative regulation of endopeptidase activity, proteolysis involved in cellular protein catabolic process and proteolysis.

## 3. Discussion

The equine tapeworm, *A. perfoliata*, remains a research neglected parasite with limited molecular information available and as such, a lack of understanding of the host–parasite interaction. Our study employed a polyomic approach to characterise adult *A. perfoliata*, generating a transcriptome of the whole worm and proteomic maps of the secretome. To our knowledge, the present study is the first to generate a de novo transcriptome assembly of this adult equine tapeworm. Moreover, we also present the first evidence that the secretome of an equine helminth parasite generates EVs, which are filled with a plethora of immune-modulatory proteins that have previously been suggested as regulators of host immune responses.

The top 50 most highly represented transcripts demonstrate the expression of common conserved genes similar to other closely related cestodes at several life stages—such as dynein light chain, tegumental protein, deoxyhypusine hydroxylase, 8 kDa glycoprotein, and expressed conserved protein—thus demonstrating the validity of the transcript assembly [62,63,64,65,66,67,68]. Moreover, the *A. perfoliata* transcriptome aligned well to 3 closely related tapeworm species, namely *H. diminuta*, *E. granulosus*, and *H. microstoma*, again confirming the validity of the dataset.

The development of the first transcriptome for *A. perfoliata* provides support to explore key proteins of importance linked to the host–parasite interface as has been demonstrated previously for other helminths [19,21,27,69]. At the host–parasite interface immune modulation is imperative for parasite survival [70,71] and consequently many immune modulatory proteins have been identified in platyhelminth species [40,45,49]. The current transcriptome and proteomic analysis has identified 454 transcripts as homologues of recognised immune modulators in other helminth species of which several are functionally expressed given their presence as part of the secretome. Notable immune modulators were identified in the top 50 most abundant proteins, such as sigma class GST (EV depleted ESP), enolase (EVs, EV surface and EV depleted ESP), calpain A (EVs, EV surface, and EV depleted ESP), and HSP90α (EVs, EV surface and EV depleted ESP). Thus, we have demonstrated the potential for an immune modulatory role of the *A. perfoliata* secretome that may have wide ranging effects on the host immune response to the parasite. The functionality of these putative immune modulators now needs further investigation.

The parasite secretome during infection is known to have an essential role in host–parasite interactions [16,26,27,28,29,30,31,32,33,34,41,72]. Our study establishes for the first time the *A. perfoliata* secretome; including proteomic analysis of in vitro secreted EVs purified through SEC, the EV surface and EV depleted ESP. We report identification of a total of 315 proteins from *A. perfoliata* whole EVs, 301 proteins from the EV surface and a further 596 proteins from EV deleted ESP. The majority of GO terms were enriched in the biological processes group, however, calcium ion binding in the molecular category was found to be the most enriched across all secretome samples, a process involved in EV biogenesis [73,74,75].

Many key secretory proteins such as GSTs, HSP90 and Enolase were secreted by *A. perfoliata* as free proteins (EV depleted ESP) during in vitro maintenance in the current study, which shows similarity to other cestodes [25,28,76,77]. Interestingly, enolase, which is described as a multifunctional protein and essential in the host immune system evasion through immunomodulation, was observed as the third-most-abundant protein in *A. perfoliata* EV depleted ESP. Likewise, enolase was found as the most abundant in *E. granulosus* [76] and *Taenia solium* ESP [25]. In addition, proteins identified in *H. diminuta* ESP, like peroxidasin, expressed conserved protein, NADP-dependent malic enzyme, deoxyhypusine hydroxylase, and particularly, basement membrane-specific heparan sulfate proteoglycan core protein, were also present in the 50 most abundant proteins in *A. perfoliata* EV depleted ESP [28].

We also confirmed for the first time that equine tapeworms, *A. perfoliata*, release whole EVs as part of the ESP, during in vitro maintenance, determined via TEM and NTA analysis. Both size and morphology of *A. perfoliata* whole EVs were similar to EVs released from other tapeworms in a spherical shaped or cup-shaped membrane surrounded by a phospholipid bilayer structure with sizes ranging from 30 to 200 nm in diameter [38,39,40,41,42,43,44]. The protein profile of *A. perfoliata* EVs also demonstrated a number of common EV markers in the top 50 abundant proteins, which have been reported in Exocarta, Vesiclepedia, and from other cestodes [38,43]. Furthermore, proteins such as H17 tegumental antigen, tegumental antigen, and tegumental protein, which were found in abundance in *A. perfoliata*, have been suggested as typical components in EVs from parasitic flatworms such as *Hymenolepis diminuta* [38], *Echinococcus granulosus* [41], and *Calicophoron daubneyi* [49].

The outer surface proteins of parasite derived EVs have crucial roles in establishing cell to cell communication, mediating cellular uptake, affecting immune recognition, and representing effector molecules [78]. The surface proteins of EVs from trematodes have been characterised for several species [47,49,79], but this is the first study to provide a proteome profile for the EV surface of a cestode species. Surface hydrolysis and gel free proteomics led to the identification of 301 surface proteins of *A. perfoliata* EVs, including many well-known EVs markers, which have been identified in other platyhelminths [47,49]. CD63 antigen, known as part of the tetraspanin family, is mainly associated with membranes of intracellular vesicles [80] and was identified on the *A. perfoliata* EV surface, although not in the top 50 most abundant proteins. Interestingly, Cathepsins (B, D, and L) which are commonly found on the surface of trematode EVs [47,49] were not observed on the *A. perfoliata* EV surface, perhaps reflecting the relative importance of parasite digestive tract secretions, that are absent from cestodes [81].

Glycolytic process and phosphorylation enzymes such as phosphoenolpyruvate carboxykinase, phosphoglycerate kinase, glyceraldehyde-3-phosphate dehydrogenase, and glucose 6 phosphate were also identified on the surface of *A. perfoliata* which are crucial enzyme activities providing a source of energy for helminths [82,83] and likely reflect the EV site of origin from the cestode tegument given *A. perfoliata* lack a digestive tract [81]. Glycogen or glucose, although sporadically available in the caecum, are likely an easy source of energy in the host’s gut which are absorbed directly through their tegument [83,84]. Additionally, membrane transport proteins on the surface of *A. perfoliata* EVs such as solute carrier family 5, plasma membrane calcium-transporting ATPase 3, band 3 anion transport protein particularly the pumps protein; ATP binding cassette subfamily B (MDR:TAP) and multidrug resistance protein were observed on the *A. perfoliata* EV surface similar to *F. hepatica* EVs [47]. These transport enzyme activities are likely to enhance the carbohydrate metabolism mechanisms in or between cells providing more nutrients and energy uptake to *A. perfoliata*.

The transcriptomics and proteomics analysis of *A. perfoliata* demonstrates the variety of key proteins that are relevant to the parasite–host interaction. We therefore further investigated three novel proteins using the transcriptome that have previously been identified as immunomodulators, namely Sigma class GSTs, HSP90α, and enolase to better understand the relationship of these proteins in *A. perfoliata* to those identified in other platyhelminths.

Sigma class GSTs have well established multi-functional roles in the host–parasite interaction, including general detoxification of xenobiotic and endogenously derived toxins and prostaglandin synthase activity and as such have been suggested as vaccine candidates within *Fasciola* [85,86,87] and *Schistosoma* [88,89,90,91,92,93,94]. We have identified two novel *A. perfoliata* Sigma class GSTs (ApGST-S1, GST-S2.1, and GST-S2.2) within the *A. perfoliata* transcriptome, confirmed by secondary structure assessment, domain analysis, and phylogenetic analysis. Furthermore, a likely functional expressed protein of ApGST-S1 was also identified in the EV depleted ESP and was initially given the protein description as AChain A, Glutathione S-transferase 28 Kda (GST class-Mu 28 kDa isozyme) following Omicsbox classification. However, domain analysis and phylogenetic analysis demonstrated that ApGST-S1 is likely be a Sigma-like GST. All *A. perfoliata* Sigma class GSTs were clustered well in the Sigma class GST clade, specifically as part of a cestode group. Moreover, the secondary characteristic structure prediction demonstrated the consistency and similarity with other cestodes and trematodes of the β-strand, α-helix, and random coils structures within the recognised Sigma class GST sequences, which are conserved regions of these proteins. Interestingly, the catalytic tyrosine residues (Y) positioned at the end of the first β-strand, which has been suggested as the key feature of the GSH binding site of the sigma class GST [52,53,54,55], was present in ApGST-S2.1 and S2.2 but missing in ApGST-S1 which was replaced with a histidine residue. However, ApGST-S1 demonstrates similarity to *H. microstoma* Sigma-like GST (Hmic; accession CDS25704) at this residue and was consequently clustered alongside this *H. microstoma* Sigma-like GST in the phylogenetic tree (bootstrap value 52%). Therefore, ApGST-S1 is likely be a Sigma-like GST with ApGST-S2.1 and S2.2 representing true Sigma class GSTs. However, given its secretion and alternative active site residue, ApGST-S1 may preferentially function as an immune modulator.

α-Enolase, also known as phosphopyruvate hydratase, is a glycolytic enzyme responsible for converting 2-phosphoglycerate (2-PG) into phosphoenolpyruvate (PEP) in the penultimate step of glycolysis [95]. Additionally, α-Enolase is also considered a multi-functional protein due to acting as a plasminogen receptor and concentrating proteolytic plasmin activity on the cell surface [96]. Functional characterisations of the *Onchocerca volvulus* α-Enolase has suggested that this enzyme possesses immunomodulatory properties due to its ability to bind to plasminogen and promote plasmin-mediated proteolysis, which subsequently leads to the degradation of the host’s extracellular matrix [37,97]. In total, three full length novel α-Enolases were identified (Apα-Enolase-1, 2, and 3) when exploring the *A. perfoliata* transcriptome with the potential of two further isoforms that were currently represented by small fragments. Following phylogenetic analysis, all three novel Apα-enolases were clustered well in the α-enolase clade, specifically as part of a cestode group. Assessing primary amino acid sequence of the translated contig hits showed Apα-enolase-1, -2, and -3 to conserve all five active site amino acid residues (His^158^, Glu^167^, Glu2^10^, Lys^343^, Lys^394^, respective positions in human α-enolase), suggesting these α-enolase *A. perfoliata* proteins would exhibit similar enzymatic activity properties commonly associated with previously characterised α-enolase proteins [58]. Apα-enolase-1, -2, and -3 also possessed similar secondary structure positioning when compared to human α-enolase (11 β-strands and 16 α-helices), which further supports the previous hypothesis. From the proteomic datasets, only Apα-enolase-1 was expressed in EVs, on the EV surface and in EV depleted ESP, Apα-enolase-2 was expressed in EV depleted ESP, suggesting key roles in host invasion/interaction. To this end, Apα-Enolase-1, -2, and -3 are confirmed to be likely functional Apα-enolase. However, the two Apα-enolase fragments lacked the five catalytic active-site residues and comparable secondary structure patterning, which suggests enzymatic activity is not conserved in these two Apα-enolase proteins and thus further investigation is required to confirm their enzymatic activity and full transcripts.

HSP90 is a molecular chaperone and a highly conserved protein involved in signal transduction, cell cycle control, stress management and folding, degradation, and transport of proteins [98,99,100,101,102,103,104]. Furthermore, HSP90 has also been thought to be involved in host immune system modulation via platyhelminth secretomes [50,105], although information on the role of HSP90 as an immune modulator in helminth infections is less extensive than that presented for sigma class GSTs and enolase. In exploring the *A. perfoliata* transcriptome, five novel HSP90s were identified (ApHSP90-1 [1.1 and 1.2], -2 [2.1, 2.2], -3, -4, and -5 [5.1 and 5.2]). All putative ApHSP90s were confirmed via secondary structure assessment and phylogenetic analysis with all five identified as the HSP90 alpha (HSP90α) isoform via phylogenetics. It has been reported that only HSP90α isoforms are secreted from cells whereas HSP90β isoforms (HSP90β) primarily operate intracellularly [106]. This is of interest to the secretome dataset, which identified HSP90α proteins in EVs, on the EV surface and in EV depleted ESP, thus supporting the importance of expression of HSP90α for production of proteins to be secreted by *A. perfoliata* and suggesting key roles in host invasion/interaction. Further work will need to be completed to elucidate if HSP90α is involved in host immune modulation in *A. perfoliata* infected horses. On further assessment of the secondary structure, only ApHSP90-4 contained the cytoplasmic HSP90 sequence motif, MEEVD, whereas ApHSP90-5 (both 5.1 and 5.2) contained KEEL which is 75% conserved to the KDEL peptide sequences of the HSP90 endoplasmic reticulum (GRP94; 94-kDa glucose-regulated protein). As expected, there were no ApHSP90 sequences that contained LKID peptide sequences, which are specific to HSP90β [56,57,99]. Three ApHSP90 sequences (ApHSP90-1.1, 1.2, 2.1, 2.2, and 3) were also shown to be closely related to other recognised cytosolic HSP90 sequences within the phylogenetic tree (accession CDS28179 HmicHSP90, CDS25067 EgraHSP90, and CDS39694 EmulHSP90). Therefore, ApHSP90-1, -2, and -3 may be the cytosolic HSP90 which are not specific to alpha (inducible isoform) or beta isoforms (constitutively expressed). The LIP and EDD peptide sequences have been suggested as a signature sequences of HSP90 alpha isoforms [57], yet they were missing in ApHSP90-4, although present in the other 4 ApHSP90. Interestingly, the IIP and EDE peptide sequences were found in ApHSP90-4 instead, which are similar to that observed in *E. granulosus* HSP90α (accession XP_024345770 and CDI70178). To this end, ApHSP90-1, -2, -3, and -4 are most likely to be a cytosolic HSP90, which of ApHSP90-4 is most likely to be HSP90α-like, whereas ApHSP90-5 is most likely to be an endoplasmic reticulum HSP90.

In the current study we have generated the first de novo transcriptome for *A. perfoliata* to support functional genomics investigations into the host–parasite interaction. In addition, the first in-depth proteomic profiles of the *A. perfoliata* secretome has been conducted to gain insights into this important interface. In doing so, we have demonstrated that the *A. perfoliata* secretome contains many proteins that have previously been identified to be involved in host–parasite interactions, namely through immune modulation of the host environment, and these are found in both EVs and EV depleted ESP. We have also identified and characterised novel potential *A. perfoliata* immune modulators, namely sigma class GST, α-enolase, and HSP90α isoforms. However, the findings demonstrate a variety of key secretory molecules from EVs and ESP, which are not limited to those characterised within the current work, yet the wider immunomodulatory activities of the *A. perfoliata* secretome need to be further investigated. Importantly, our study demonstrates that *A. perfoliata* does have the potential to modulate the horse host immune response.

## 4. Materials and Methods

### 4.1. Collection of Adult A. perfoliata and In Vitro Maintenance

Live adult *A. perfoliata* were collected from the caecum at the ileocecal valve of naturally infected horses immediately post-slaughter from a commercial abattoir. Specimens were washed thrice in pre-warmed sterile phosphate-buffered saline (PBS; pH 7.4; Thermo Scientific, Loughborough, UK) at 39 °C to remove host contamination. For subsequent RNA extraction, six *A. perfoliata* from six separate infections were immediately snap-frozen in liquid nitrogen for 1 min and stored on dry ice for transportation to the laboratory, where they were stored at −80 °C until RNA extraction. For secretome proteomic analysis, 50 live adult *A. perfoliata* per replicate were maintained in vitro from three individual horse infections following the method previously described [107]. Briefly, *A. perfoliata* were maintained at 39 °C for 5 h in Dulbecco’s modified Eagle’s medium (DMEM, Gibco, Thermo Scientific, Loughborough, UK; supplemented with 2.2 mM Calcium acetate, 2.7 mM Magnesium sulphate, 61 mM glucose, 15 mM HEPES pH 7.0–7.6, gentamycin (5 μg/mL), and 1 μM serotonin). Following the maintenance period, adult *A. perfoliata* and precipitated debris were removed. The culture supernatant was collected and immediately stored at −80 °C until extracellular vesicle purification and further proteomics analysis.

### 4.2. Total RNA Extraction and Purification

Total RNA was extracted and purified from adult *A. perfoliata* (*n* = 6) using the Direct-zol™ RNA MiniPrep Plus Kit (Zymo Research, Cambridge, UK). *A. perfoliata* were removed from −80 °C and ≤50 mg of the soma, including the scolex, was removed and cut into small pieces before transferring into a 2 mL microcentrifuge tube containing 600 μL RNA Isolation Reagent, TRI Reagent^®^ (Zymo Research, Cambridge, UK). Tissue samples were subsequently disrupted via bead beating by adding a pre-frozen (−80 °C) 5 mm stainless-steel bead (Qiagen, Manchester, UK) and samples placed in a TissueLyser LT (Qiagen, Manchester, UK) for 2 min at 50 oscillations per s. The bead beating process was repeated where tissue disruption was not complete. Samples were centrifuged at 15,000× *g* for 30 s to pellet any remaining debris and the supernatant extracted following the manufacturer’s protocol. RNA concentration was determined using a NanoDrop1000 spectrophotometer (Thermo Scientific, Loughborough, UK) and the integrity determined via a 2100 Bioanalyzer (Agilent Technologies, CA, USA) assessment, following the manufacturer’s instructions.

### 4.3. RNA-Seq Library Construction and Next Generation Sequencing

Purified RNA from all samples were sequenced at the Translation Genomics facility in IBERS, Aberystwyth University. Briefly, RNA purity was assessed using Qubit^®^ RNA HS Assay Kits with the Qubit^®^ Fluorometer (Invitrogen, Thermo Scientific, Loughborough, UK). cDNA libraries were then constructed by reverse transcribing 500 ng of total RNA from each sample using the TruSeq RNA Library Preparation Kit v2 according to the Low Sample (LS) Workflow (Illumina, Cambridge, UK). RNA adapter Indexes were added and ligated on both ends of cDNA to allow multiple indexing of samples pooled together. The cDNA fragments with adapters then underwent PCR amplification (Illumina, Cambridge, UK). cDNA quality was determined on a 1.2% *w*/*v* agarose gel.

Amplified cDNA libraries were quantified using an Ultrospec EPOCH (BioTek, China), at an absorbance measurement of 280 nm to normalise a pooling volume of each sample library prior to sequencing. The final concentration of pooled cDNA libraries was quantified using a Qubit^®^ 2.0, dsDNA BR Assay Kit (Invitrogen, Life Technologies, Paisley, UK). Cluster generation and sequencing were performed according to the MiSeq Workflow using MiSeq Reagent Kit v3 (Illumina, Cambridge, UK). Briefly, cDNA libraries were adjusted in equimolar concentration to 10 nM concentration with 10 nM Tris HCl (Melford Laboratories, Suffolk, UK) and 0.05 % *v*/*v* Tween-20 solution (Sigma-Aldrich, Merck Life Sciences, Dorset, UK) followed by diluting to 2 nM with buffer EB (Qiagen, Manchester, UK). cDNA libraries were denatured to a single stranded DNA using 0.1 M sodium hydroxide (Sigma-Aldrich, Merck Life Sciences, Dorset, UK) and diluted again to final loading concentration at 6 pM in hybridisation buffer (Illumina, Cambridge, UK). Samples were clustered onto a MiSeq flow cell and paired-end sequenced on a Miseq™, according to standard protocols (Illumina, Cambridge, UK). Base pairs (bp) per read were generated in 2 × 75 bp format.

### 4.4. De Novo Transcriptome Sequencing Analysis Pipeline

The sequencing pipeline was performed through the Galaxy web platform hosted by IBERS, Aberystwyth University (version 17.01; [108,109,110]). Prior to assembly, all raw FASTQ sequencing data files were assessed via FastQC (Galaxy tool version 0.69; Babraham Bioinformatics; [111]). All reads with a phred quality scores <20 were discarded, although no reads were found below this cut-off. Based on the FastQC assessment, reads were trimmed via Trimmomatic (Galaxy Version 0.36.0; [112]). Illuminaclip was initially used followed by Slidingwindow to remove from the 3′ end and Minlen to remove any reads below 36 bp long. Trimmed reads were again assessed through FastQC to ensure that the read quality of the new RNA-Seq datasets had phred scores of ≥30 across more than 70% of the bases.

De novo assembly of reads was completed in Trinity (v2.11.0; [113,114]) using default parameters. To determine a common set of transcripts between all six biological replicates, all six assemblies were clustered together with Cluster Database at High Identity with Tolerance (cd-hit) software (version 4.8.1; [115,116,117,118,119]). All coding regions within transcript sequences were identified using the assembled unitranscripts as input through Transdecoder (part of the Trinity package; [114]). To detect open reading frames (ORF), parameter settings used at least 100 amino acid long and the ORFs retention of 3000.

Host (horse) contamination was removed from the transcriptome by comparing to protein and CDS files from the *Equus caballus* genome from Ensemble (version 3.0; [120]), and *Hymenolepis microstoma* as the closest relative genome sequenced cestode (PRJEB124; [64]), using BLASTp or BLASTx with default options and a minimum e-value of 0.1. Transcripts which were more similar to the host (horse) rather than to *Hymenolepis microstoma* were deemed to be likely host contaminants and were subsequently removed from the transcriptome.

### 4.5. Functional Annotation and Gene Ontology (GO) Terms Analysis

The resulting assembly was functionally annotated using Omicsbox [121] to predict the functional description (DE) and GO functional classification of the unigenes. The expression level of transcripts using RNA-seq data was quantified by Salmon [122]. The top 50 abundant transcripts were searched against the Omicsbox output to obtain protein descriptions. Transcripts not found in the output were subsequently translated into protein sequences using ExPASy Translate tools [123] followed by manually BLASTp against the NCBI (nr) protein database using a protein query (BLASTp; [124]) and cut-off set at 1.0 × 10^−03^.

### 4.6. Bioinformatic Analysis of Potential Immune Modulators

The transcriptome was analysed for the presence of characterised immune modulators, previously identified in helminths by performing tBLASTn searched against the *A. perfoliata* transcriptome through BioEdit Sequence Alignment Editor (Version 7.2.6.1; [125]). The number of expected hits of similar quality (e-value) cutoff was set at 1.0 × 10^−15^ using various immunomodulator bait peptide sequences retrieved from Genbank and NCBI Reference Sequence (see immunomodulators listed in Appendix A). Subsequently, the top 5 hits from each bait sequence were searched and translated with ExPASy Translate tools [123] to identify the best opening reading frames (ORFs). The peptide sequence of the bait proteins were submitted to Pfam database (version 34.0; [126]) to confirm for protein domain conservation.

Three potential immune modulators in the transcript database, sigma class glutathione transferase (Sigma class GST), cytoplasmic heat shock protein 90 (HSP90), and alpha-enolase (α-Enolase), were selected for further bioinformatic analysis. Protein sequences of recognised Sigma class GST, HSP90 family (alpha and beta isoforms) and α-Enolase from 13, 22, 14 different species, respectively, covering mammalians, nematodes, trematodes, and cestodes were retrieved from Genbank and NCBI Reference Sequence (see recognised Sigma class GST, HSP90 and α-Enolase proteins sequences in Appendix A, [52,56,64,127,128,129,130,131,132,133,134,135,136,137,138,139,140,141,142]. Recognised sequences were blasted against the *A. perfoliata* transcriptome (cutoff set at 1.0 × 10^−15^, 1.0 × 10^−20^, and 1.0 × 10^−15^ for Sigma GST, HSP90, and α-Enolase, respectively) and representative *A. perfoliata* sequences translated into protein sequences, as previously described. To ensure that selected protein sequences were either Sigma class GST, HSP90 or α-Enolase, each protein sequence was searched against the NCBI (nr) protein database using a protein query (BLASTp; [124]). Subsequently, all representative protein sequences were classified into protein super-families, domain prediction and functional site analysis through InterProScan databases (version 77.0; [143,144]). The resulting InterPro domains classified as Sigma class GST, HSP90, and α-enolase with N-terminal domain (NTD), C-terminal domain (CTD) were kept as a unique sigma class GST, HSP90, and α-enolase protein sequence. All unique classified sigma class GST, HSP90, and α-enolase sequences, or one representative if isoforms were presented, were taken as a final sigma class GST, HSP90, and α-enolase representative protein sequence in *A. perfoliata* for subsequent phylogenetic analysis.

### 4.7. Sequence Alignment and Phylogenetic Analysis of Potential Immune Modulators

All multiple sequence alignments of the resulting final sigma class GST, HSP90α, and α-enolase representative protein sequences of *A. perfoliata* and recognised sigma class GST, HSP90α, and α-enolase protein sequences were completed using ClustalW through BioEdit Sequence Alignment Editor. The secondary characteristic structure including beta sheets and alpha helixes of novel *A. perfoliata* sigma class GST, HSP90α, and α-enolase sequences were then predicted using the Predict Secondary Structure (PSIPRED) Protein Analysis Workbench (PSIPRED 4.0; [145,146]) followed by protein domain identification and architecture analysis using Simple Modular Architecture Research (SMART) tools [147,148] to obtain the novel *A. perfoliata* Sigma class GST, HSP90α, and α-enolase sequences.

Subsequently, phylogenetic trees were constructed and visualised in MEGA X (version 10.1.7; [149,150]). Reliability of the phylogenetic tree was estimated with 1000 bootstrap replicates, using both a neighbour-joining (NJ) method and a maximum likelihood (ML) method. For NJ method, the parameters were set as the correction of the amino acid data based on the gamma distribution of rates at 1.0 with a Poisson correction method, pairwise deletions and number of treat at 3. For ML method the parameters were set with a likelihood of amino acid data determined based upon five discrete gamma rate categories. An initial tree for the heuristic search was obtained automatically by applying NJ and BIONJ algorithms to a matrix of pairwise distances estimated using a Jones–Taylor–Thornton (JTT) substitution model, ML heuristic method, with nearest-neighbor-interchange (NNI), and number of threads at 3.

### 4.8. Extracellular Vesicles Purification by Size Exclusion Chromatography

EVs were purified from *A. perfoliata* culture media following the protocol described [45]. Briefly, media was centrifuged at 4 °C at 300× *g* for 10 min and then 700× *g* for 30 min. Subsequently, residual cells and debris were removed by filtering supernatant through a 0.45 μm PES syringe membrane filter (STARLAB, Milton Keynes, UK). *A. perfoliata* supernatant was concentrated using 10 KDaMWCO Amicon^®^ Ultra-15 Centrifugal Filter Units (MerckMillipore, Merck Life Sciences, Dorset, UK), following the manufacturer’s guidelines. Briefly, samples were centrifuged at 3000× *g* for 20 min at 4 °C, until approximately 500 μL of sample was retained in the filter unit. Filtration flow-through was stored at −80 °C for further analysis.

*A. perfoliata* EVs were then purified using a qEV original size exclusion chromatography (SEC) column (iZON Science, Oxford, UK), according to the manufacturer’s protocol. Briefly, 10 mL of filtered (0.22 μm, STARLAB) PBS (pH 7.4) was loaded through the qEVoriginal SEC column, followed by 500 μL of the concentrated supernatant. The first 3 mL of the filtration flow-through was discarded and SEC purified EVs collected from the next 1.5 mL of the filtration flow-through. Subsequently, 10 mL of filtered PBS was added to the qEVoriginal SEC column and the next 7.5 mL of the filtration flow-through collected EV depleted SEC ESP. Both EVs and EV depleted SEC ESP were stored at −80 °C until further proteomics analysis.

### 4.9. Characterization of Extracellular Vesicles Released from A. perfoliata

#### 4.9.1. Transmission Electron Microscopy (TEM) Analysis

*A. perfoliata* EV sample (*n* = 3) preparation for TEM was performed following the protocol described [151]. Briefly, 10 μL of EVs in 2% paraformaldehyde (PFA) were fixed onto formvar/Carbon coated Copper TEM grids (400 Mesh, Agar Scientific, Stansted, UK). Following fixation, each grid was then washed in 100 μL of PBS (pH 7.4) for 1 min followed by fixing in a 1% (*v*/*v*) glutaraldehyde solution (Sigma-Aldrich, Merck Life Sciences, Dorset, UK) for 5 min. Each TEM grid was washed with distilled water for 2 min for a total of eight times. Grids were then contrast-stained in 50 μL of uranyl-oxalate solution (pH 7) for 5 min. Finally, TEM grids were embedded in 50 μL of methyl cellulose uranyl-oxalate, for 10 min on ice, and stored at the room temperature before imaging via transmission electron microscope (JEM1010 Transmission Electron Microscope, Jeol, Tokyo, Japan) at 80 kV as previously described [45]. The size (dimension) of 200 EVs per purification sample imaged by TEM were measured using ImageJ (version 1.52a; [152].

#### 4.9.2. Nanoparticle Tracking Analysis (NTA)

*A. perfoliata* EVs (*n* = 3) underwent nanoparticle tracking with size distribution and number of particles in each replicate determined using a Nanosight NS500 system (Malvern Instruments, Malvern, UK) equipped with a green 532 nm laser and a high sensitivity electron multiplying charge-coupled device (EMCCD) camera (Andor Technology, Belfast, UK), following the manufacturer’s instruction. Samples were diluted in PBS (pH 7.4) to obtain a concentration of particles ranging between 10^6^ and 10^9^ particles/mL (Malvern Instruments, Malvern, UK). For each sample, videos of the particles moving under Brownian motion were captured, with a camera level of 15. Subsequently, the captured video data were analysed using the NanoSight software (NTA version 3.2 Dev Build 3.2.16) to assess the particle size and concentration of EVs, with the analysis setting set at a detection threshold of 5.

#### 4.9.3. Extracellular Vesicle Surface Protein Hydrolysis

Surface proteins of SEC purified EVs of *A. perfoliata* were removed through hydrolysis with trypsin as previously described [49]. Briefly, SEC purified EVs were diluted with PBS to a final concentration of 200 μg in 250 μL total volume. Sequencing grade modified trypsin (100 μg/mL; Roche, U.K) was added to the EVs obtained a final concentration of 50 μg/mL and incubated for 5 min at 37 °C. The treated EVs were then centrifuged for 1 h at 100,000× *g* at 4 °C. The resulting supernatant was stored at −20 °C prior to gel free mass spectrometry analysis.

### 4.10. Secretome Proteomics Analysis

#### 4.10.1. One Dimensional Sodium Dodecyl Sulfate Polyacrylamide Gel Electrophoresis

EV depleted SEC ESP were concentrated by precipitation with ice-cold 20% (*w*/*v*) trichloroacetic acid (Thermo Scientific, Loughborough, UK) in 100% acetone as previously described [107]. Precipitated pellets were re-suspended in Isoelectric Focusing (IEF) Buffer Z (8 M urea, 2% *w*/*v* CHAPS (C_32_H_58_N_2_O_7_S), 33 mM dithiothreitol, 0.5% carrier ampholytes *v*/*v* BioLyte^®^ 3/10) prior to protein quantification. EVs depleted ESP samples were quantified using Bradford assay [153] according to the manufacturer’s protocol, through an UV–visible spectrophotometer (Cary 50, Agilent Technologies, Cheshire, UK) at an absorbance measurement of A595 nm. The Qubit^®^ Protein Assay Kits along with the Qubit^®^ 2.0 Fluorometer (Invitrogen, Thermo Scientific, Loughborough, UK) was employed to quantify the concentration of EV samples according to the manufacturer’s protocol. Samples were evaporated under a vacuum centrifugation for approximately 1 h to concentrate the samples, which was repeated until an acceptable concentration was reached and quantified. All EVs and EV depleted ESP samples were stored at −20 °C until further 1D SDS-PAGE electrophoresis.

Both EVs and EV depleted ESP samples were run on a mini 1D SDS-PAGE gel, 7cm 12.5 % resolving acrylamide gels. All samples were mixed with 4x SDS loading buffer, heated to 95 °C for 10 min and centrifuged at 21,000× *g* for 10 min. All samples were run at a constant voltage of 70V (BioRad) for approximately 20 min until the bromophenol blue moved through the stacking gel, and then increased to 150 V until completion. Gels were fixed in 40% (*v*/*v*) ethanol and 10% (*v*/*v*) acetic acid (Thermo Scientific) for 1 h and stained overnight with colloidal Coomassie™ Brilliant Blue (80% (*v*/*v*) dye stock solution and 20% (*v*/*v*) methanol). Gels were de-stained with 1% *v*/*v* acetic acid and then imaged with a GS-800™ Calibrated Densitometer (BioRad).

#### 4.10.2. Trypsin In-Gel Digestion and Liquid Chromatography-Tandem Mass Spectrometry

SDS PAGE lanes containing either EVs or EV depleted ESP were divided into 9 and 12 sections respectively. Each of these sections were excised for in-gel digestion with trypsin, as previously described [31]. Briefly, excised gel bands were de-stained with 50% (*v*/*v*) acetonitrile (and 50% (*v*/*v*) 50 mM ammonium bicarbonate (Thermo Scientific) at 37 °C for 15 min). The supernatant was discarded and the process repeated until gel pieces were de-stained. Gel bands were then dehydrated in 100% acetonitrile at 37 °C for 15 min, and the gel dried at 50 °C. Following drying, 10 mM dithiothreitol in 50 mM ammonium bicarbonate was added to gel pieces and incubated at 80 °C for 30 min before then incubating with 55 mM iodoacetamide in in 50 mM ammonium bicarbonate (Sigma-Aldrich) at room temperature for 20 min. Gel pieces were washed with 50% (*v*/*v*) acetonitrile and 50% (*v*/*v*) 50 mM ammonium bicarbonate at room temperature for 15 min, dehydrated with 100% acetonitrile at room temperature for 15 min and then dried at 50 °C. Gel pieces were rehydrated and digested with 50 mM ammonium bicarbonate containing trypsin at 10 ng/μL at 37 °C for approximately 16 h. Gel pieces were then centrifuged at 10,000× *g* for 10 min and then MilliQ water added before placing on a shaker at room temperature for 10 min, and the supernatant retained. 50% (*v*/*v*) acetonitrile and 5% (*v*/*v*) formic acid were again added to the gel pieces at room temperature for 60 min. Gel pieces were centrifuged as previously and the supernatant retained and added to the previously retained extraction. All retained supernatants containing peptides were dried until pelleted ready for mass spectrometry analysis.

Liquid chromatography tandem mass spectrometry was performed at the Advanced Mass Spectrometry Facility, School of Biosciences, University of Birmingham as a commercial service. Briefly, dried peptide pellets were re-suspended in 0.1% *v*/*v* formic acid and then loaded with an autosampler to be analysed by liquid chromatography tandem mass spectrometry (Q Exactive™ HF Hybrid Quadrupole-Orbitrap™ Mass Spectrometer, Thermo Scientific) equipped with a TriVersa Nanomate (Advion, Harlow, UK) and nanoflow liquid chromatography system (Dionex, Thermo Scientific).

#### 4.10.3. Protein Identification and Gene Ontology Terms Enrichment Analysis

Protein identification of *A. perfoliata* proteomics profiles were performed through MASCOT [154] hosted by IBERS, Aberystwyth University, according to the method described [107]. Briefly, the acquired MS/MS spectra (Mascot Generic Files) were submitted to a MASCOT MS/MS ions search (Matrix Science, v2.6; [154]) against the *A. perfoliata* transcriptome and *Equus caballus* genome version 3.0. Search parameters used the following: trypsin enzymatic cleavage with up to one missed cleavage allowed, fixed modifications to carboxymethyl cysteine with variable modifications set for oxidation of methionine, fixing fragment monoisotopic mass error with peptide tolerances of ±1.2 Da and MS/MS of ±0.6 Da, peptide charge 1+, 2+, and 3+, monoisotopic, data format with mascot generic, electrospray ionization (ESI) TRAP. The resulting identified proteins that indicated the identity or extensive homology (*p* < 0.05) were selected according to the individual MASCOT ions score with scores set at greater than 48 for EVs and EVs depleted ESP samples and 47 for EVs surface samples. Subsequently, unique peptides presented in at least two replicates were then used for searching against the *A. perfoliata* annotation database (obtained from the Omicsbox) to assign the protein description and Gene Ontology (GO) terms. The resulting number of proteins identified from MS/MS analysis within *A. perfoliata* proteomics datasets were then visualised in Venn-diagrams using InteractiVenn [155]. Gene Ontology terms enrichment analysis (GOEA) on gene sets of all proteomics datasets was performed using GOATOOLS python package (v0.5.9, [156]) whether the GO terms were propagated up the hierarchy (prop) or were not propagated up the hierarchy (nop) (*p* < 0.05 identified significance).

## Figures and Tables

**Figure 1 pathogens-10-00912-f001:**
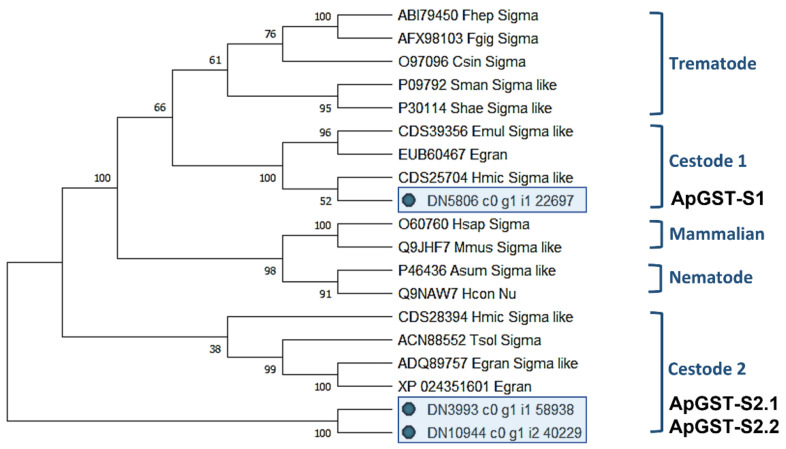
Maximum likelihood (ML) tree with JTT matrix-based model inferred from *A. perfoliata* Sigma class GST amino acid sequences. The bootstrap consensus tree inferred from 1000 replicates. Initial tree(s) for the heuristic search were obtained automatically by applying Neighbor-Join and BioNJ algorithms to a matrix of pairwise distances estimated using a JTT model, and then selecting the topology with superior log likelihood value. A discrete Gamma distribution was used to model evolutionary rate differences among sites (five categories (+G, parameter = 4.2179)). This analysis involved 19 amino acid sequences. There was a total of 279 positions in the final dataset. Evolutionary analyses were conducted in MEGA X.

**Figure 2 pathogens-10-00912-f002:**
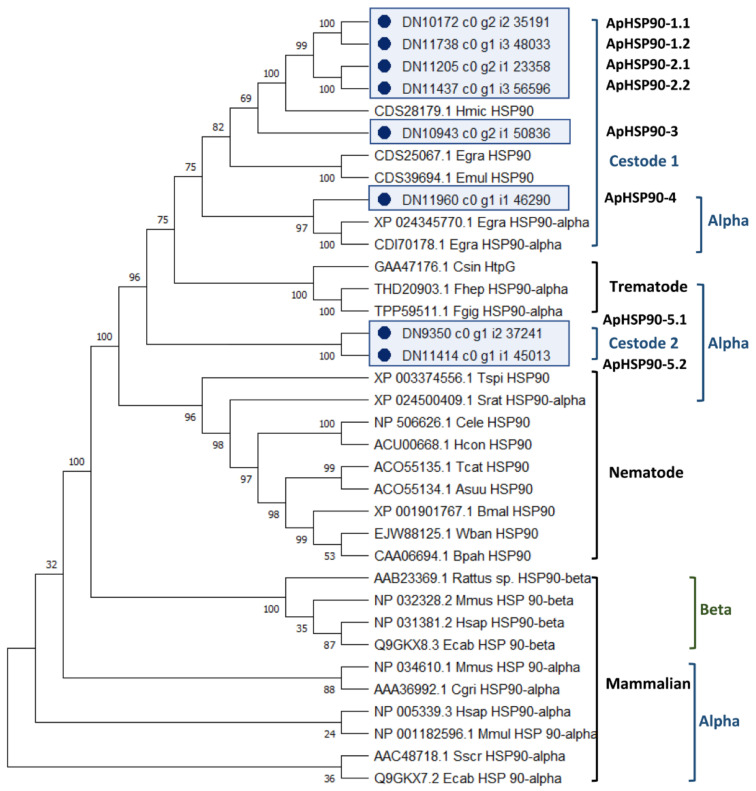
The phylogenetic tree of *A. perfoilata* Heat Shock Protein 90 inferred by using Maximum likelihood (ML) method and JTT matrix-based model. The bootstrap consensus tree inferred from 1000 replicates. The tree with the highest log likelihood (-16485.50) is shown. Initial tree(s) for the heuristic search were obtained automatically by applying Neighbor-Join and BioNJ algorithms to a matrix of pairwise distances estimated using a JTT model, and then selecting the topology with superior log likelihood value. A discrete Gamma distribution was used to model evolutionary rate differences among sites (five categories (+G, parameter = 1.4964)). This analysis involved 35 amino acid sequences. There was a total of 885 positions in the final dataset. Evolutionary analyses were conducted in MEGA X.

**Figure 3 pathogens-10-00912-f003:**
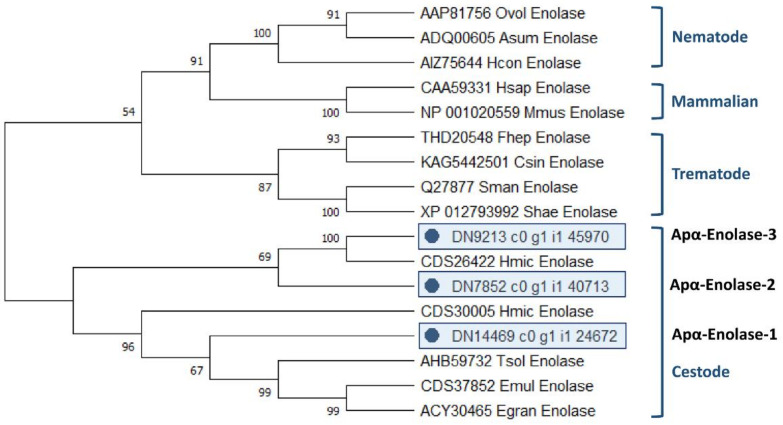
Maximum likelihood (ML) tree with JTT matrix-based model inferred from *A. perfoliata* α-Enolase amino acid sequences. The bootstrap consensus tree inferred from 1000 replicates. Initial tree(s) for the heuristic search were obtained automatically by applying Neighbor-Join and BioNJ algorithms to a matrix of pairwise distances estimated using a JTT model, and then selecting the topology with superior log likelihood value. A discrete Gamma distribution was used to model evolutionary rate differences among sites (five categories (+G, parameter = 0.4274)). This analysis involved 19 amino acid sequences. There were a total of 597 positions in the final dataset. Evolutionary analyses were conducted in MEGA X.

**Figure 4 pathogens-10-00912-f004:**
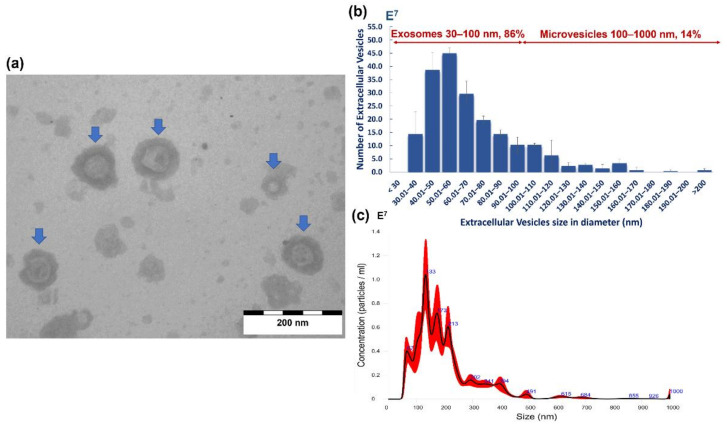
Characterisation of the in vitro maintenance *A. perfoliata* extracellular vesicles (EVs) by transmission electron microscopy (TEM) and nanoparticle tracking analysis (NTA): (**a**) The TEM image from purified *A. perfoliata* derived EVs at 80 kV; scale bar is 200 nm; demonstrates that *A. perfoliata* secreted extracellular vesicles (blue arrows). *A. perfoliata* EV morphology is in a spherical shaped membrane surrounded by a phospholipid bilayer structure; (**b**) A mean size distribution of the isolated size exclusion chromatography (SEC) purified *A. perfoliata* EVs (mean ± SE; 200 EVs per replicates; *n* = 3). The number of exosomes is shown in the size range of 30–100 nm (86%) and microvesicles in the size range of 100–1000 nm (14%); (**c**) Representative histogram showing the EV particle size distribution and average finite track length adjustment (FTLA) concentration (EVs × 10^7^/mL vs size in nm) of SEC purified *A. perfoliata* EVs at 1:600 dilution with main peaks at approximately 67–213 nm determined by nanoparticle tracking analysis. Averaged FTLA concentration, as red areas, specify the standard deviation (SD) between measurements and blue numbers indicate the maxima of individual peaks.

**Figure 5 pathogens-10-00912-f005:**
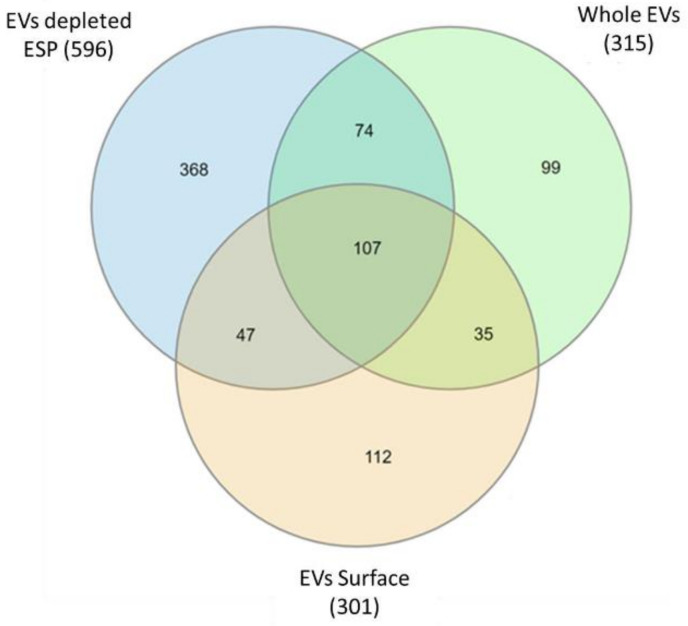
Venn diagrams comparing the proteins identified in *Anoplocephala perfoliata* whole extracellular vesicle (EVs), extracellular vesicle (EV) surface, and extracellular vesicle depleted excretory-secretory products (EV depleted ESP) retrieved from mass spectrometry analysis and MASCOT via MS/MS Ion Search against the *A. perfoliata* transcriptome.

**Table 1 pathogens-10-00912-t001:** Summary statistics of the trimmed Illumina sequencing, de novo transcriptome assembly and TransDecoder protein sequences of adult *A. perfoliata* from naturally infected horses (*n* = 6).

**Illumina RNA Sequencing**	**Trimmed Reads**
Total reads (bp)	104,519,050
Mean (SD) reads per sample (bp)	2,903,307 (608,363)
Sequence length (bp)	36–76
GC percentage (%)	46
**De Novo Trinity Assembly**	**Assembled Transcript**
Total assembled length (bp)	56,913,324
Number of contigs	74,607
Number of contigs (without Isoforms)	26,653
Mean (SD) contig lengths per sample (bp)	763 (762)
Max. contig lengths (bp)	11,266
Min. contig lengths (bp)	201
Contigs N50	1155
**TransDecoder**	**Peptide Dataset**
Number of protein sequences	34,341

**Table 2 pathogens-10-00912-t002:** The top 50 most abundant transcripts in *A. perfoliata* calculated from Salmon and expressed as a TPM value. The description of each transcript is demonstrated from Omicsbox functional annotation.

Rank	Gene ID	Length	Mean Effective Length	Mean Number Reads	Mean TPM	Description
**1**	DN10805_c3_g6_i1_11702	1036	858	407,936	46,601	Uncharacterised
**2**	DN10805_c3_g2_i9_11698	1754	1591	264,055	16,600	Uncharacterised
**3**	DN6887_c0_g1_i1_76818	384	212	34,703	16,001	Uncharacterised
**4**	DN8498_c0_g2_i2_20529	333	162	17,101	10,383	Uncharacterised
**5**	DN14755_c0_g1_i1_41815	314	148	10,387	6916	Uncharacterised
**6**	DN8064_c0_g1_i1_68336	477	315	13,860	4289	Uncharacterised
**7**	DN8427_c0_g1_i1_72770	410	234	10,236	4283	Uncharacterised
**8**	DN7932_c0_g1_i1_6806	401	225	9709	4239	Uncharacterised
**9**	DN12386_c1_g1_i9_39644	4006	3843	158,692	4199	Transcript antisense to ribosomal rna protein
**10**	DN10805_c3_g2_i7_11692	3260	3097	133,004	4199	Uncharacterised
**11**	DN3788_c0_g1_i1_50567	426	265	10,398	3869	Uncharacterised
**12**	DN8606_c1_g1_i3_340	314	146	5533	3826	Uncharacterised
**13**	DN12640_c1_g1_i8_71952	9547	9384	316,400	3293	Uncharacterised
**14**	DN10987_c0_g1_i3_65072	874	711	23,587	3277	Expressed conserved protein
**15**	DN5384_c0_g1_i1_75533	414	241	8058	3248	Uncharacterised
**16**	DN8805_c0_g1_i1_60212	444	283	8962	3129	Uncharacterised
**17**	DN12068_c0_g3_i1_63804	1028	851	25,275	2996	Expressed conserved protein
**18**	DN13746_c0_g1_i1_26674	358	199	5871	2919	Uncharacterised
**19**	DN10069_c0_g1_i3_2055	985	809	21,491	2662	Expressed conserved protein
**20**	DN11085_c1_g1_i5_42186	381	221	5786	2641	Uncharacterised
**21**	DN9799_c0_g1_i1_46050	508	346	9173	2617	Dynein light chain 1, cytoplasmic
**22**	DN118_c0_g1_i1_52814	595	420	10,683	2540	Tegumental protein
**23**	DN10418_c3_g5_i1_34715	483	311	8002	2501	Immunogenic protein
**24**	DN6144_c0_g1_i1_49576	641	469	11,504	2425	Dynein light chain type 1 2
**25**	DN3712_c0_g1_i1_698	602	440	10,689	2400	Dynein light chain 1, putative
**26**	DN10763_c1_g5_i1_22341	281	128	3136	2298	Uncharacterised
**27**	DN7817_c0_g1_i1_51569	1013	850	19,568	2191	Expressed conserved protein
**28**	DN9590_c0_g2_i1_51137	480	318	6844	2115	Uncharacterised
**29**	DN10367_c0_g1_i3_29401	1522	1347	28,911	2057	Expressed conserved protein
**30**	DN3791_c0_g1_i1_14014	632	470	9552	2007	Uncharacterised
**31**	DN4055_c0_g1_i1_61482	518	346	7092	1989	Dynein light chain type 1 2
**32**	DN6146_c0_g1_i2_38481	514	352	6601	1906	Uncharacterised
**33**	DN15763_c0_g1_i1_57495	831	668	12,973	1887	Tegumental protein
**34**	DN10204_c0_g1_i2_51272	516	354	6655	1828	Uncharacterised
**35**	DN12126_c0_g1_i9_41589	2283	2120	38,789	1826	Uncharacterised
**36**	DN502_c0_g1_i1_32021	462	284	5326	1809	Uncharacterised
**37**	DN10922_c1_g1_i5_14239	1292	1129	19,933	1778	Deoxyhypusine hydroxylase
**38**	DN5442_c0_g1_i1_26423	439	261	4666	1728	Uncharacterised
**39**	DN5954_c0_g1_i2_54752	370	201	3606	1703	Uncharacterised
**40**	DN9433_c0_g1_i5_18544	424	251	4344	1674	8 kDa glycoprotein
**41**	DN12201_c0_g3_i1_43360	334	176	3135	1672	No hit
**42**	DN11009_c0_g1_i1_3069	711	535	9247	1669	Uncharacterised
**43**	DN11588_c0_g1_i4_75797	406	245	4072	1669	Uncharacterised
**44**	DN10667_c1_g1_i2_72481	469	296	5181	1666	Uncharacterised
**45**	DN5960_c0_g1_i2_30732	274	122	1928	1571	Uncharacterised
**46**	DN7341_c0_g1_i1_31615	484	322	4974	1532	Dynein light chain 1, cytoplasmic
**47**	DN1202_c0_g1_i1_62670	642	470	7386	1532	Profilin allergen
**48**	DN10641_c0_g1_i1_57690	585	423	6391	1528	Uncharacterised
**49**	DN15681_c0_g1_i1_62555	578	416	6427	1513	Uncharacterised
**50**	DN9547_c0_g1_i1_1108	381	221	3260	1477	Uncharacterised

**Table 3 pathogens-10-00912-t003:** Summary statistics of nano-particle tracking analysis of size exclusion chromatography purified *Anoplocephala perfoliata* EVs.

Parameters	Mean EV ± SE
Mean (nm)	199.1 ± 5.3
Mode (nm)	144.7 ± 7.5
SD (nm)	108.7 ± 8.5
D10 (nm)	105.9 ± 2.6
D50 (nm)	168.3 ± 4.6
D90 (nm)	337.9 ± 14.7
Concentration (particles/mL)	9.42 × 10^11^ ± 7.20 × 10^11^
Concentration (particles/frame) 1:600 dilution	81.2 ± 6.1
Concentration (centres/frame) 1:600 dilution	116.1 ± 7.1

**Table 4 pathogens-10-00912-t004:** The top 50 most abundant proteins putatively identified in *A. perfoliata* EVs proteomics dataset by 1D SDS-PAGE, LC MS/MS (GeLC) and a MASCOT search at the significance threshold score above 48. Protein descriptions were given from Omicsbox. Protein hits shaded in grey represent known helminth released immune modulators, as also identified in the *A. perfoliata* transcriptome.

No.	Protein Description	Sequence ID	Number of Sequenced Peptides	MASCOT Score
**1**	WD repeat and FYVE domain-containing protein 3	DN11838_c0_g1_i2_56054	78	1933
**2**	Actin, cytoplasmic type 5	DN10334_c0_g1_i3_17117	41	958
**3**	Myosin XV	DN10026_c0_g1_i2_27828	36	346
**4**	Myosin heavy chain 10 or non-muscle myosin IIB	DN10438_c0_g2_i2_34624	34	1235
**5**	Leucine-rich repeat-containing protein	DN11834_c0_g1_i11_56034	30	611
**6**	Otoferlin	DN10250_c0_g1_i1_26991	29	245
**7**	Fascin 2	DN10161_c0_g1_i1_59685	29	599
**8**	Calpain A	DN9786_c0_g1_i1_72692	27	1113
**9**	Annexin A7	DN8700_c0_g1_i1_600	26	747
**10**	Phosphoenolpyruvate carboxykinase	DN11364_c0_g2_i1_39228	25	691
**11**	Actin, cytoplasmic type 5	DN10334_c0_g1_i1_17115	24	865
**12**	Heat shock 70 kDa protein 4	DN12581_c0_g1_i2_73130	23	440
**13**	Expressed conserved protein	DN11921_c1_g2_i1_46303	22	400
**14**	Von Willebrand factor A domain containing protein	DN11931_c2_g1_i10_58523	22	396
**15**	Expressed conserved protein	DN7822_c0_g2_i1_27391	22	131
**16**	Annexin A7	DN7793_c0_g1_i1_23754	21	321
**17**	Aldo keto reductase family 1-member B4	DN11165_c0_g1_i1_24177	21	416
**18**	Expressed conserved protein	DN10367_c0_g1_i3_29401	21	473
**19**	Solute carrier family 5	DN12278_c0_g6_i1_68615	20	642
**20**	Tegumental antigen	DN5781_c0_g1_i1_2927	20	674
**21**	Programmed cell death 6- interacting protein	DN10491_c0_g3_i2_21343	19	303
**22**	Peroxidasin	DN10163_c0_g1_i1_35077	18	419
**23**	Enolase	DN14469_c0_g1_i1_24672 (Apα-Enolase-1)	18	447
**24**	Ubiquitin-60S ribosomal protein L40	DN12547_c0_g1_i1_46361	18	335
**25**	Annexin A13 (Annexin XIII)	DN9930_c0_g1_i1_67885	17	1023
**26**	Tegumental protein	DN15763_c0_g1_i1_57495	17	260
**27**	Molecular chaperone HtpG/Heat shock protein 90 alpha	DN11960_c0_g1_i1_46290 (ApHSP90-4)	16	239
**28**	Expressed conserved protein	DN8957_c0_g1_i1_66134	16	299
**29**	Glycoprotein Antigen 5	DN9013_c0_g1_i2_47406	16	476
**30**	Annexin A13 (Annexin XIII)	DN12676_c0_g1_i9_72045	15	480
**31**	Alpha 2 macroglobulin	DN12789_c0_g1_i5_70580	15	293
**32**	Phosphoglycerate kinase	DN11218_c0_g1_i1_75075	14	338
**33**	Annexin B9-like isoform X1	DN11220_c0_g1_i12_22997	14	650
**34**	Non-lysosomal glucosylceramidase	DN9975_c0_g1_i9_4448	14	274
**35**	Solute carrier family 5	DN10836_c0_g1_i4_11677	14	600
**36**	Basement membrane-specific heparan sulfate proteoglycan core protein	DN9818_c0_g2_i1_37822	14	237
**37**	Tegumental protein	DN118_c0_g1_i1_52814	14	417
**38**	H17g protein tegumental antigen	DN11977_c0_g1_i2_72857	14	357
**39**	Hypothetical transcript	DN9865_c0_g1_i1_63028	14	594
**40**	Cytosolic malate dehydrogenase	DN10181_c0_g1_i1_47181	13	139
**41**	Putative anoctamin	DN11493_c0_g1_i2_56859	13	145
**42**	Plasma membrane calcium- transporting ATPase 3	DN11817_c3_g5_i1_69816	13	234
**43**	Uncharacterised	DN6547_c0_g1_i3_66219	13	250
**44**	Annexin A13 (Annexin XIII)	DN12676_c0_g1_i5_72043	13	547
**45**	Von Willebrand factor A domain containing protein	DN10879_c1_g1_i8_12057	13	381
**46**	Annexin A4-like	DN12342_c0_g1_i2_39792	13	824
**47**	Carbonic anhydrase	DN11803_c0_g3_i1_69783	13	255
**48**	Annexin A7	DN11263_c0_g1_i4_60744	12	320
**49**	Calpain	DN4288_c0_g1_i1_31793	12	321
**50**	Unnamed protein product, partial	DN11248_c0_g2_i1_22893	12	267

**Table 5 pathogens-10-00912-t005:** The top 50 most abundant proteins putatively identified on the *A. Perfoliata* EV surface proteomic dataset by gel-free LC MS/MS and a MASCOT search at the significance threshold score above 47. Protein descriptions were given from Omicsbox. Protein hits shaded in grey represent known helminth released immune modulators, as also identified in the *A. perfoliata* transcriptome.

No.	Protein Description	Sequence ID	Number of Sequenced Peptides	MASCOT Score
**1**	WD repeat and FYVE domain-containing protein 3	DN11838_c0_g1_i2_56054	72	1302
**2**	Expressed conserved protein	DN10367_c0_g1_i3_29401	50	1039
**3**	Myosin heavy chain 10 or non-muscle myosin IIB	DN10438_c0_g2_i2_34624	50	1372
**4**	P29	DN11822_c0_g2_i2_55872	44	1617
**5**	Basement membrane-specific heparan sulfate proteoglycan core protein	DN9818_c0_g2_i1_37822	44	470
**6**	Spectrin alpha chain	DN11694_c0_g1_i1_54338	37	686
**7**	Expressed conserved protein	DN11921_c1_g2_i1_46303	37	701
**8**	Myosin XV	DN10026_c0_g1_i2_27828	34	296
**9**	Collagen alpha-2(I) chain	DN6173_c0_g1_i4_63619	33	759
**10**	Expressed conserved protein	DN7822_c0_g2_i1_27391	32	114
**11**	Expressed conserved protein	DN10746_c0_g1_i6_22331	31	841
**12**	Annexin A7	DN8700_c0_g1_i1_600	30	1130
**13**	Leucine-rich repeat-containing protein	DN11834_c0_g1_i3_56030	29	596
**14**	Annexin A13 (Annexin XIII)	DN9930_c0_g1_i1_67885	29	1313
**15**	Microtubule actin cross linking factor 1	DN10747_c0_g1_i5_35870	28	231
**16**	Peroxidasin	DN10163_c0_g1_i1_35077	28	393
**17**	Spectrin alpha actinin	DN11195_c0_g3_i1_24267	28	231
**18**	Myosin heavy chain	DN11757_c0_g1_i1_61366	28	613
**19**	Heat shock 70 kDa protein 4	DN12581_c0_g1_i2_73130	27	634
**20**	Calpain A	DN9786_c0_g1_i1_72692	26	931
**21**	Expressed conserved protein	DN11614_c0_g2_i3_53973	24	338
**22**	Plasma membrane calcium-transporting ATPase 3	DN10463_c3_g1_i2_34420	23	329
**23**	Von Willebrand factor A domain containing protein	DN10879_c1_g1_i8_12057	23	310
**24**	Enolase	DN14469_c0_g1_i1_24672 (Apα-Enolase-1)	23	474
**25**	Calpain	DN4288_c0_g1_i1_31793	23	333
**26**	Galectin carbohydrate recognition domain	DN6894_c0_g1_i2_12735	21	631
**27**	Tegumental antigen	DN5781_c0_g1_i1_2927	21	1112
**28**	No hit	DN10801_c0_g1_i14_11633	20	594
**29**	Phosphoenolpyruvate carboxykinase	DN11364_c0_g2_i1_39228	20	253
**30**	Molecular chaperone HtpG/Heat shock protein 90 alpha	DN11960_c0_g1_i1_46290 (ApHSP90-4)	20	372
**31**	Annexin A7	DN7793_c0_g1_i1_23754	20	336
**32**	Glycoprotein Antigen 5	DN9013_c0_g1_i2_47406	20	458
**33**	Annexin A7	DN11263_c0_g1_i4_60744	20	355
**34**	H17g protein tegumental antigen	DN11977_c0_g1_i2_72857	19	470
**35**	Programmed cell death 6-interacting protein	DN10491_c0_g3_i2_21343	19	645
**36**	Expressed conserved protein	DN12262_c0_g1_i1_68587	18	203
**37**	Actin modulator protein	DN8972_c0_g1_i1_2323	17	395
**38**	Otoferlin	DN10250_c0_g1_i1_26991	17	527
**39**	Ornithine aminotransferase	DN9481_c0_g1_i1_55273	17	289
**40**	Unnamed protein product	DN12187_c0_g1_i1_66436	17	335
**41**	Fascin 2	DN10161_c0_g1_i1_59685	16	535
**42**	Actin, cytoplasmic type 5	DN10334_c0_g1_i3_17117	16	570
**43**	Calmodulin	DN5211_c0_g1_i2_35332	16	381
**44**	Expressed conserved protein	DN8957_c0_g1_i1_66134	16	681
**45**	Paramyosin	DN10354_c0_g1_i1_3720	15	299
**46**	Serine/threonine kinase	DN8156_c0_g1_i2_52113	15	212
**47**	Protein kinase C and casein kinase substrate in neurons protein 1	DN7152_c0_g1_i2_8970	15	456
**48**	Tegumental protein	DN118_c0_g1_i1_52814	14	205
**49**	Lysyl oxidase	DN7852_c0_g1_i1_1626	14	189
**50**	Phosphoglycerate kinase	DN11218_c0_g1_i1_75075	14	123

**Table 6 pathogens-10-00912-t006:** The top 50 most abundant proteins putatively identified in *A. perfoliata* EV depleted ESP proteomic datasets by 1D SDS-PAGE, LC MS/MS, and a MASCOT search at the significance threshold score above 48. Protein descriptions were given from Omicsbox. Protein hits shaded in grey represent known helminth released immune modulators, as also identified in the *A. perfoliata* transcriptome.

No.	Protein Description	Sequence ID	Number of Sequenced Peptides	MASCOT Score
**1**	WD repeat and FYVE domain-containing protein 3	DN11838_c0_g1_i2_56054	292	6943
**2**	Basement membrane-specific heparan sulfate proteoglycan core protein	DN9818_c0_g2_i1_37822	169	2575
**3**	Enolase	DN14469_c0_g1_i1_24672 (Apα-Enolase-1)	146	5177
**4**	Alpha 2 macroglobulin	DN12789_c0_g1_i5_70580	118	2655
**5**	Ornithine aminotransferase	DN9481_c0_g1_i1_55273	106	2044
**6**	Aldo keto reductase family 1-member B4	DN10754_c1_g2_i7_35857	96	3057
**7**	Deoxyhypusine hydroxylase	DN10922_c1_g1_i5_14239	95	1993
**8**	Protein disulfide-isomerase	DN9431_c0_g1_i1_5402	91	3042
**9**	Peroxidasin	DN10163_c0_g1_i1_35077	89	2110
**10**	Cytosolic malate dehydrogenase	DN10181_c0_g1_i1_47181	75	1053
**11**	Heat shock 70 kDa protein 4	DN12581_c0_g1_i2_73130	74	1512
**12**	Actin, cytoplasmic type 5	DN10334_c0_g1_i3_17117	68	1765
**13**	Glycogen phosphorylase	DN9054_c0_g2_i1_35634	63	865
**14**	Fascin 2	DN10161_c0_g1_i1_59685	62	952
**15**	Lysosomal alpha-glucosidase	DN10704_c0_g1_i4_9857	60	932
**16**	Gynecophoral canal protein	DN2510_c0_g1_i1_41860	59	991
**17**	Phosphoenolpyruvate carboxykinase	DN11364_c0_g2_i1_39228	57	1088
**18**	Protein disulfide-isomerase A3	DN6375_c0_g1_i1_15550	55	1134
**19**	Von Willebrand factor A domain containing protein	DN10879_c1_g1_i8_12057	53	1406
**20**	Calpain A	DN9786_c0_g1_i1_72692	52	1568
**21**	Spectrin alpha chain	DN11694_c0_g1_i1_54338	50	513
**22**	Putative zinc binding dehydrogenase	DN10593_c0_g1_i11_28936	50	1031
**23**	Phosphoglycerate kinase	DN11218_c0_g1_i1_75075	50	958
**24**	Fructose-bisphosphate aldolase	DN10221_c0_g1_i1_65390	46	909
**25**	NADP-dependent malic enzyme	DN8932_c0_g1_i2_2337	45	1210
**26**	Actin modulator protein	DN8953_c0_g1_i6_52210	45	977
**27**	EF hand family protein	DN9944_c0_g2_i4_42845	42	1195
**28**	Expressed conserved protein	DN11614_c0_g2_i3_53973	41	201
**29**	Molecular chaperone HtpG/ Heat shock protein 90 alpha	DN11960_c0_g1_i1_46290 (ApHSP90-4)	41	801
**30**	Spectrin alpha actinin	DN11195_c0_g3_i1_24267	40	467
**31**	Basement membrane-specific heparan sulfate proteoglycan core protein	DN9714_c0_g1_i3_20481	40	770
**32**	Beta galactosidase	DN10618_c0_g1_i1_32785	39	752
**33**	Transketolase	DN9107_c0_g1_i1_32351	39	985
**34**	Glucose-6-phosphate isomerase	DN10660_c0_g1_i2_45795	38	709
**35**	Puromycin sensitive aminopeptidase	DN10270_c0_g1_i2_26817	37	706
**36**	Calsyntenin 1	DN10458_c0_g1_i1_34309	37	638
**37**	Gynecophoral canal protein	DN7995_c0_g1_i1_6847	37	653
**38**	Glycerol kinase	DN8664_c0_g1_i1_13160	37	1104
**39**	Peptidyl-glycine alpha-amidating monooxygenase A	DN8251_c0_g1_i1_16141	36	725
**40**	Hypothetical transcript	DN9865_c0_g1_i1_63028	36	642
**41**	Adenylosuccinate synthetase	DN10697_c0_g3_i1_6876	36	810
**42**	Glucose-6-phosphate 1-dehydrogenase	DN11811_c2_g4_i1_55908	35	1378
**43**	Myosin heavy chain 10 or non-muscle myosin IIB	DN12309_c0_g1_i3_64783	35	70
**44**	Expressed conserved protein	DN11119_c0_g1_i2_24053	34	662
**45**	Putative actin-interacting protein 1	DN1602_c0_g1_i1_69974	34	218
**46**	Expressed conserved protein	DN7822_c0_g2_i1_27391	34	595
**47**	Puromycin sensitive aminopeptidase	DN10270_c0_g1_i1_26816	33	488
**48**	Phosphoglucomutase	DN12341_c1_g3_i1_64791	33	449
**49**	Ubiquitin modifier activating enzyme 1	DN11247_c1_g1_i4_75101	33	784
**50**	Peptidyl prolyl cis trans isomerase B	DN4872_c0_g1_i1_43079	32	202

## Data Availability

The Transcriptome Shotgun Assembly project has been deposited at DDBJ/EMBL/GenBank under the accession GJFT00000000. The version described in this paper is the first version, GJFT01000000. The *A. perfoliata* transcriptome is available for BLAST analysis at https://sequenceserver.ibers.aber.ac.uk. Proteomics data from LC-MSMS analysis has been deposited to the ProteomeXchange Consortium via the PRIDE partner repository with the dataset identifier PXD027105 and 10.6019/PXD027105.

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
