# Peer review of "Evidence of Immune Modulators in the Secretome of the Equine Tapeworm Anoplocephala perfoliata"

_pathogens, 2021, doi:10.3390/pathogens10070912_

Round 1
Reviewer 1 Report
Manuscript is well written and has provided very useful information for the basic science research. Authors have to make sure all the scientific names and other latin words are italicized (This is lacking all through out the methods and results).
Author Response
We thank the reviewer for their kind comments. We apologise for the lack of italics. This was intially correct but has reverted to non-italics during the submission process. All species names are now in italics as requested. Thanks again.
Reviewer 2 Report
Wititkornkul and colleagues performed a detailed molecular analysis on A. perfoliata, performing RNA sequencing and generating the first A. perfoliata transcriptome. In addition, the authors also analysed the excretory secretory products (ESP) from A. perfoliate, in particular, free proteins and extracellular vesicles (EVs). The authors presented an extensive and detailed description of several potential molecular targets on host -parasite interaction. The present work is well written and easy to follow and can be accepted for publication in its present form.
Author Response
We thank the reviewer for their time to read our work and for their very kind comments. Best wishes.
Reviewer 3 Report
This comprehensive work on Anoplocephala sp. tapeworms of horses provides several valuable datasets for those in the field. The annotated transcriptome, initial data on secreted protein molecules, and on vesicle secretion are extremely valuable to subsequent development of novel vaccines, diagnostic assays, and treatments for this parasite. The methodology is sound, and the manuscript well-written. I have no significant critiques. I commend the authors on such a strong submission.
Author Response
Many thanks to the reviewer for their time to read this paper and for their extremely positive comments. Best wishes for the future.